# The Assessment of Subregions in the Frontal Lobe May Be Feasible in the Differential Diagnosis of Progressive Supranuclear Palsy—Parkinsonism Predominant (PSP-P) and Multiple System Atrophy (MSA)

**DOI:** 10.3390/diagnostics12102421

**Published:** 2022-10-07

**Authors:** Piotr Alster, Natalia Madetko-Alster, Bartosz Migda, Michał Nieciecki, Dariusz Koziorowski, Leszek Królicki

**Affiliations:** 1Department of Neurology, Medical University of Warsaw, 03-242 Warsaw, Poland; 2Diagnostic Ultrasound Lab, Department of Pediatric Radiology, Medical Faculty, Medical University of Warsaw, 03-242 Warsaw, Poland; 3Department of Nuclear Medicine, Children’s Memorial Health Institute, 04-736 Warsaw, Poland; 4Department of Nuclear Medicine, Medical University of Warsaw, 02-097 Warsaw, Poland

**Keywords:** progressive supranuclear palsy, PSP, frontal perfusion, SPECT

## Abstract

Progressive Supranuclear Palsy—Parkinsonism Predominant (PSP-P) is associated with moderate responsiveness to levodopa treatment and a possible lack of typical PSP milestones. The clinical manifestation of PSP-P poses difficulties in neurological examination. In the early stages it is often misdiagnosed as Parkinson’s Disease, and in the more advanced stages PSP-P shows more symptoms in common with Multiple System Atrophy—Parkinsonian type (MSA-P). The small number of tools enabling differential diagnosis of PSP-P and MSA leads to the necessity of searching for parameters facilitating in vivo diagnosis. In this study, 14 patients with PSP-P and 21 patients with MSA-P were evaluated using Single Photon Emission Computed Tomography. Considering the fact that PSP is linked with frontal deficits, regions of the frontal lobe were assessed in the context of hypoperfusion and their possible usefulness in the differential diagnosis with MSA-P. The outcome of the work revealed that the right middle frontal gyrus was the region most significantly affected in PSP-P.

## 1. Introduction

Progressive Supranuclear Palsy—Parkinsonism Predominant (PSP-P) and Multiple System Atrophy—Parkinsonian type (MSA-P) are clinical manifestations of atypical Parkinsonisms. Due to their overlapping symptomatology, searching for feasible methods enabling efficient examination seems crucial. Patients diagnosed with PSP-P do not always show typical known milestones of Progressive Supranuclear Palsy (PSP). PSP-P, a subcortical phenotype of PSP, is associated with longer life expectancy than cortical PSP and Progressive Supranuclear Palsy—Richardson Syndrome (PSP-RS) [1]. Moreover, the clinical manifestation is affected by levodopa responsiveness and symmetrical tremors may be confusing in interpretation [2]. Earlier studies concerning the role of neuroimaging in PSP-P and MSA-P identified Magnetic Resonance Imaging (MRI) and Single Photon Emission Computed Tomography (SPECT) as possible methods of assessment [3,4,5,6,7,8,9]. Perfusion SPECT revealed significant differences between PSP-P and MSA-P in the frontal lobe, with more pronounced hypoperfusion in this region observed in PSP-P. However, the previous analyses did not include extended evaluation of the subregions of the frontal lobe.

Frontal atrophy is a commonly described feature of PSP. The atrophy in PSP is evolving in the anterior to posterior direction. It is initiated in the insula and subsequently passes to the frontal lobe, and eventually to the temporal, parietal, and occipital lobes [10]. Most of the works refer to PSP without indicating subtypes, or are based on the examination of patients with PSP-RS. A study comparing abnormalities of brain metabolism in PSP and MSA revealed hypometabolism in the cerebellum and putamen in MSA and in the medial prefrontal cortices, nucleus caudatus, frontal cortices, and mesencephalon in PSP [11]. A different work presented decreased frontal and midbrain glucose metabolism as the neuroimaging features of PSP [12]. The regions impacted by hypoperfusion and hypometabolism partly overlap the ones significantly affected by tau inclusions in PSP [13]. A study concerning evaluation of PSP-P and PSP-RS revealed a more pronounced frontal lobe tau pathology in PSP-RS when compared to PSP-P. Interestingly, no correlations were found between the level of atrophy and the significance of tau pathology [14]. Frontal atrophy accompanied by pronounced deficits in neuropsychological assessment, e.g., Frontal Assessment Battery (FAB), are interpreted as additional tools facilitating examination [4]. Deficits associated with abnormalities in the frontal lobe of patients with PSP are related to verbal fluency deficits [15]. In PSP-P phonemic verbal fluency was not found to be beneficial in early diagnosis of PSP-P [16]. The frontal deficits assessed using FAB in PSP-P and PSP-RS revealed that FAB is more feasible in the examination of PSP-RS than in PSP-P. In this work the authors intended to determine whether certain parts of the frontal lobe are more relevant to differential diagnosis.

## 2. Methods

### 2.1. Material

The research was conducted in the Department of Neurology of the Medical University of Warsaw and Department of Nuclear Medicine of the Mazovian Brodno Hospital between January 2017 and December 2019. In total, 21 patients (14 females, 7 males) with a clinical diagnosis of MSA-P and 14 patients (8 females, 6 males) with a clinical diagnosis of PSP-P, aged 50–80, were included in the study. The duration of the disease varied from 3 to 6 years. The levodopa dose varied between 0 and 1000 mg. The patients were all right-handed. All of the patients included in the study gave their written consent. The authors excluded patients with neoplasms, multiple chronic vascular changes or patients who had previously suffered a stroke. The clinical diagnosis was based on the criteria of diagnosis of MSA and PSP [17,18]. The research was undertaken by physicians experienced in examining movement disorders.

### 2.2. SPECT

The evaluation using SPECT was carried out at the Department of Nuclear Medicine at Mazovian Brodno Hospital. The analysis of perfusion was conducted using the same method as preceding studies by the research group. Evaluation of cerebral blood flow was conducted using technetium-99m hexamethylpropyleneamine oxime ([^99m^Tc] Tc-HMPAO). Patients received an amount of 740 mBq of [^99m^Tc] Tc-HMPAO in a silent, dimly lit room. The possession administration was conducted in a supine position with a SPECT/CT scan (Symbia T6, Siemens) on a dual-head gamma camera with a low-energy high-resolution parallel-hole collimator. A step-and-shoot acquisition mode was used. Sequences of 128 frames on a 128 × 128 matrix were utilized (64 projections per head, 30 s per projection). The photopeak was set at 140 keV with a 10% window either way. Iterative restoration (eight iterations, eight subsets, 7 mm Gauss filter), scatter correction and CT attenuation correction were completed. Post-processing evaluation was performed using Scenium software (Siemens Medical Solutions USA, Inc., Malvern, PA, USA). The SPECT regions of interest (ROIs) (including subregion ROIs) were initially baselined using Scenium software (an essential component of the Siemens workstation) based on T1-weighted MRI images of a standard brain dataset. The segmentation of the specific subregions were performed by using MR examinations. The shape and size of the SPECT-evaluated brains were calibrated in accordance with the shape and size of the standard brains from the dataset. The baselined ROIs were then used to predict expected ROIs of the SPECT images of the examined brains. Finally, total maximum and minimum counts were automatically analyzed in each ROI of the investigated brain SPECT scans and were compared using Scenium with measurements derived from the standard brain SPECT scan datasets. No control group was evaluated as part of this research, instead the data were referred to a reference database comprising the [^99m^Tc] Tc-HMPAO brain scans of 20 healthy volunteers aged 64–86 years old (males and females). All comparisons were automatically presented as standard deviations using Scenium. The values of standard deviations from ROIs were assessed in various locations in the brain by statistical analysis. The outcome of the research was interpreted by a specialist experienced in nuclear medicine.

### 2.3. Statistical Analysis

Gathered data were analyzed using Statistica software (version 13.1 Dell. Inc. Statsoft). Data distribution was assessed with the Shapiro–Wilk test. Due to non-normal distribution all parameters are expressed as medians with lower (Q1) and upper (Q3) quartile and their interquartile range (Q1–Q3). For group comparison we have used the U Mann–Whitney test. Significant results are presented as box plots (Figure 1). For a final decision in regard to statistical significance we have used a corrected *p*-value after Bonferroni correction to control the False Discovery Rate (FDR). A calculated *p*-value of 0.0025 was considered significant.

## 3. Results

Table 1 presents detailed values of SPECT perfusion in regard to analyzed parts of the frontal lobe with given median values, lower and upper quartiles with their interquartile range (Q1–Q3) for the whole group, as well as for subgroups of MSA-P and PSP-P patients. The highest difference between median values of absolute SPECT perfusion between MSA-P and PSP-P patients was observed in the right middle frontal gyrus (MSA-P = 1.1 vs. PSP-P = −2.9). This difference was statistically significant (*p* < 0.0025, Table 1, Image 1). Additionally, high differences between median values in the analyzed subgroups were observed for left and right inferior frontal opercular, left and right inferior frontal triangular, right precentral gyrus, right superior frontal gyrus dorsolateral, left and right superior frontal gyrus medial, left superior frontal medial orbital with *p* < 0.05, but higher than the corrected *p* = 0.0025 (Table 1, Figure 2 and Figure 3).

## 4. Discussion

The results suggested the significance of detailed analysis of the frontal lobe neuroimaging in the examination of PSP-P. The issue concerning the differentiation of PSP-P and MSA-P has been relatively poorly described. The two entities, though based on different pathologies, are problematic in their differential diagnosis. PSP-P and MSA-P may manifest clinical features which overlap with Parkinsonian syndrome, with a possibly moderate response to levodopa treatment, as well as preserved or benignly deteriorated cognitive abilities. The neuroimaging parameters introduced as Magnetic Resonance Parkinsonism Index 2.0 (MRPI 2.0) to examine PSP-P seem to be feasible in its differentiation from PD, but are not sufficiently specific in comparison to MSA-P [5,7,8,9]. Previous work by this research group found that mesencephalon/pons ratio and MRPI may be beneficial in the neuroimaging differentiation of PSP-P and MSA-P [5]. The role of MRPI in the examination of PSP-P and MSA-P was also found to be distinguishing the diseases in a separate study [3]. The hypoperfusion of the frontal lobe was also described, but lacked discrimination of crucial regions [6]. Other work describing PSP-P and MSA-P showed increased iron accumulation in subcortical nuclei of the brain in both diseases [19].

The research is affected by several limitations. It was conducted on relatively small groups (14–21 patients), but it is based on rare diseases—in the context of PSP-P it is related with the evaluation of a phenotype associated with up to 35% cases of PSP [20]. Due to the fact that the patients were alive during the study, no neuropathological examinations were performed. No additional control group was included in the study, but the software used in the study enabled comparison with a group of age-matched healthy volunteers.

The analysis of the frontal regions of interest seems to be an evolving issue in the in vivo examination of PSP-P. Further research requires more analyses of the medial frontal lobe of the non-dominant hemispheres, likely based on more specific radiotracers. More analyses highlighting the significance of the deterioration in dominant and non-dominant frontal lobes of patients with PSP would enrich the overview of the pathophysiology of the diseases.

## Figures and Tables

**Figure 1 diagnostics-12-02421-f001:**
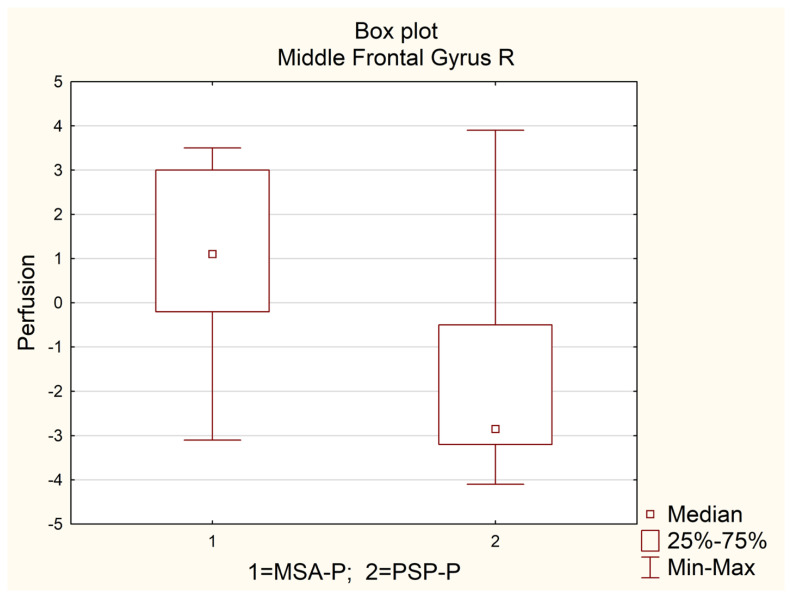
Box-plot presenting differences in the perfusion of PSP-P and MSA-P.

**Figure 2 diagnostics-12-02421-f002:**
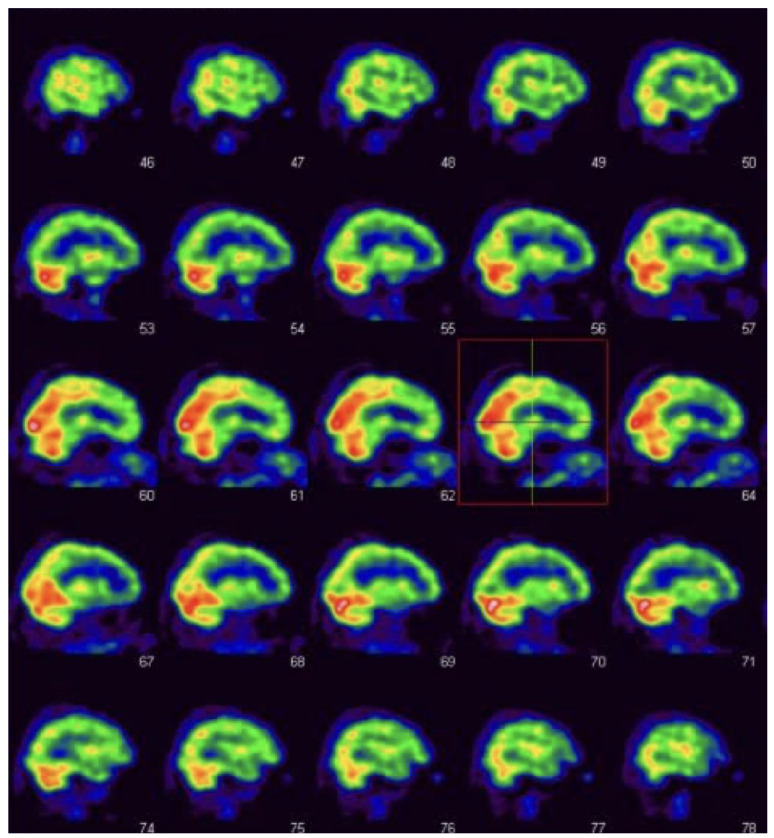
Frontal hypoperfusion in PSP-P—sagittal plane. Red Box: Pronounced decrease of perfusion within the middle frontal gyrus.

**Figure 3 diagnostics-12-02421-f003:**
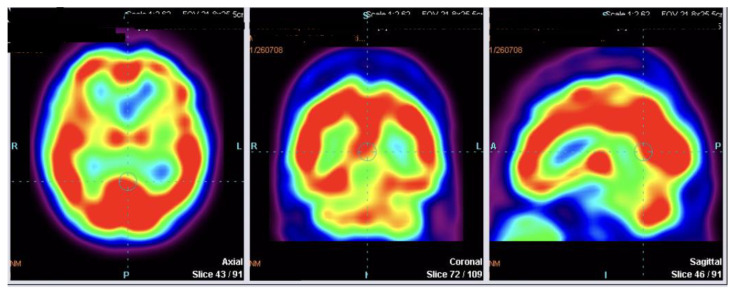
Lack of frontal hypoperfusion in MSA-P—axial, coronal, and sagittal plane.

**Table 1 diagnostics-12-02421-t001:** Descriptive statistics and group comparison.

	Total N = 35				MSA-P (N = 21)				PSP-P (N = 14)				*p*
	Median	Lower Quartile	Upper Quartile	Quartile Range	Median	Lower Quartile	Upper Quartile	Quartile Range	Median	Lower Quartile	Upper Quartile	Quartile Range	
INFERIOR FRONTAL OPERCULAR L	−0.5	−2.4	0.8	3.2	0.0	−0.5	1.4	1.9	−2.4	−3.5	−1.2	2.3	0.0038
INFERIOR FRONTAL OPERCULAR R	0.5	−2.8	2.3	5.1	1.4	0.2	2.7	2.5	−2.4	−3.4	0.3	3.7	0.0047
INFERIOR FRONTAL ORBITAL L	−0.9	−2.7	0.2	2.9	−0.4	−2.2	0.5	2.7	−2.3	−3.3	−0.3	3.0	0.0617
INFERIOR FRONTAL ORBITAL R	−0.1	−1.7	0.9	2.6	0.6	−1.1	1.5	2.6	−1.4	−2.5	−0.1	2.4	0.0529
INFERIOR FRONTAL TRIANGULAR L	−0.8	−2.8	1.4	4.2	0.2	−1.0	1.4	2.4	−2.7	−3.7	−0.8	2.9	0.0184
INFERIOR FRONTAL TRIANGULAR R	0.6	−2.5	2.8	5.3	1.6	0.1	3.0	2.9	−2.1	−3.7	1.9	5.6	0.0384
MIDDLE FRONTAL GYRUS L	−0.5	−1.4	1.3	2.7	0.1	−0.8	1.7	2.5	−0.8	−2.7	1.2	3.9	0.1524
MIDDLE FRONTAL GYRUS R	0.0	−2.9	1.6	4.5	1.1	−0.2	3.0	3.2	−2.9	−3.2	−0.5	2.7	0.0013
MIDDLE FRONTAL GYRUS ORBITAL PART L	−0.8	−2.7	0.1	2.8	−0.4	−1.5	0.1	1.6	−1.3	−3.4	−0.1	3.3	0.2386
MIDDLE FRONTAL GYRUS ORBITAL R	−1.7	−3.0	−0.4	2.6	−1.4	−2.5	−0.7	1.8	−2.3	−3.3	−0.2	3.1	0.5333
PRECENTRAL GYRUS L	−0.5	−1.3	0.5	1.8	−0.2	−1.3	1.2	2.5	−0.8	−1.2	−0.1	1.1	0.2523
PRECENTRAL GYRUS R	0.6	−0.7	1.7	2.4	1.0	0.4	2.0	1.6	−0.2	−1.3	0.6	1.9	0.0211
SUPERIOR FRONTAL GYRUS DORSOLATERAL L	−0.1	−1.3	1.5	2.8	−0.1	−0.8	1.5	2.3	−1.0	−2.0	1.0	3.0	0.2966
SUPERIOR FRONTAL GYRUS DORSOLATERAL R	0.0	−1.1	1.0	2.1	0.3	−0.1	1.2	1.3	−1.1	−2.3	0.5	2.8	0.0220
SUPERIOR FRONTAL GYRUS MEDIAL L	0.4	−0.6	1.0	1.6	0.6	−0.2	1.3	1.5	−0.7	−1.7	0.6	2.3	0.0182
SUPERIOR FRONTAL GYRUS MEDIAL R	0.2	−1.5	1.3	2.8	0.3	0.2	1.3	1.1	−1.5	−2.6	0.0	2.6	0.0116
SUPERIOR FRONTAL MEDIAL ORBITAL L	0.1	−1.5	1.4	2.9	0.9	−0.5	1.7	2.2	−0.8	−2.9	0.4	3.3	0.0489
SUPERIOR FRONTAL MEDIAL ORBITAL R	0.2	−1.8	1.1	2.9	0.7	−0.7	1.2	1.9	−1.3	−2.6	0.6	3.2	0.0529
SUPERIOR FRONTAL GYRUS ORBITAL L	−0.1	−1.9	0.8	2.7	0.0	−1.0	0.8	1.8	−0.3	−1.9	0.3	2.2	0.3905
SUPERIOR FRONTAL GYRUS ORBITAL R	−1.0	−2.2	0.1	2.3	−0.8	−1.5	0.3	1.8	−1.6	−2.7	0.0	2.7	0.2067

Legend: Me = median; Q1 = lower quartile; Q3 = upper quartile; Q1–Q3 = quartile range; *p* = *p* value for U Mann–Whitney test.

## Data Availability

The data are available on request.

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
