# Peer review of "The Assessment of Subregions in the Frontal Lobe May Be Feasible in the Differential Diagnosis of Progressive Supranuclear Palsy—Parkinsonism Predominant (PSP-P) and Multiple System Atrophy (MSA)"

_diagnostics, 2022, doi:10.3390/diagnostics12102421_

Round 1

Reviewer 1 Report

The authors assessed SPECT perfusion in parts of the frontal lobe to differentiate between PSP-P and MSA-P patients. They found that the right middle frontal gyrus showed the most significantly different signal in the two patient cohorts. This is of importance for clinicians and researchers working in the field of movement disorders. Regarding the number of patients included, I think the submission as a brief report ist feasable.

I have one major suggestion: 

- Inclusion of clinical data (i.e., disease duration, levodopa use, ...) would aid in replication of the research and derivation of its clinical impact.

Moreover, some minor points need to be noted: 

- I would suggest combining Table 1 and Table 2.

- A legend for hypoperfusion values for Image 1 would be feasable.

- The first paragraph of the discussion might be more feasable in the introduction part of the manuscript.

- Regarding reference 3: As the text of this reference is written in Japanese, a similar one in english would be more feasable to use.

Author Response

Dear Reviewer #1,

Authors are grateful for all of the comments, which improved our manuscript. We have revised the work accordingly to the suggestions:

  1. Inclusion of clinical data (i.e., disease duration, levodopa use, ...) would aid in replication of the research and derivation of its clinical impact.

The information was added.

  1. I would suggest combining Table 1 and Table 2.

The change was implemented.

  1. A legend for hypoperfusion values for Image 1 would be feasible.

The values for hypoperfusion were provided in Table 1.

  1. The first paragraph of the discussion might be more feasible in the introduction part of the manuscript.

The change was implemented.

  1. Regarding reference 3: As the text of this reference is written in Japanese, a similar one in english would be more feasable to use.

The other references regarding PSP-P and MSA-P differentiation were mentioned in the 5. and 6th position in the references. As the authors intended to acknowledge all of the works directly associated with the issue of PSP-P and MSA-P, we believe that maintaining this position despite the fact it is written in Japanese is important.

Best regards,

Piotr Alster

Reviewer 2 Report

Introduction

Authors should describe what PSP-RS is since it is mentioned for the first time.

Materials

2.1 Methods

Authors should specify if patients of the two groups are males, females or both as they done for the control group later.

2.2 Spect

1. Authors should write spect in capital letters

2. The possession administration was conducted in a supine position with….

2.3 Statistical Analysis

- Author should explain why they are saying that “Significant results are presented as box plots” but there is no box plot in the paper.

-

Table 1.

-Authors should adjust the format of the table. They can use a legend in the didascalie such as M = Median, LQ = Lower Quartile, etc. or Q1, Q3. Q3-Q1. Probably is better to add the p value of Table2 in the Table 1. It’s very difficult to correlate the data between the two tables.

- Authors should not highlight in red anything; they should argue the result in the text

Image 1 (should be Figure 1)

Authors should explain why is the image cut of and why there is a square only just around the pic n 63. Even more they should put also an image of the Frontal Hypoperfusion in MSA-P. It is advisable to describe the figure in more detail.

Results

-Authors should explain why they put the data related to the whole group but they never show data related to the control group.

-Authors should better explain how they heve done the analysis. They wrote “The highest difference between median values of absolute SPECT perfusion between MSA-P and PSP-P patients was observed in the right middle frontal gyrus (MSA-P = 1.1 vs PSP-P = -2.9)”. Do they used the U Mann-Whitney test between which values? How the control group was used in the analysis??

Discussion

-Authors should replace PSP-Richardson Syndrome by inserting the acronym PSP-RS as it will be described as PSP-Richardson Syndrome in the introduction.

-Generally, authors should enter acronyms or the complete name the first time they introduce a concept.

See: Frontal Assessment Battery, MR, ROIs

-Author should replace (5,7,8,9) with (5,7-9)

-Authors should insert the reference when they are saying “Previous work by this research group found that mesencephalon/pons ratio and MRPI may be beneficial in the neuroimaging differentiation of PSP-P and MSA-P”.

-Authors should explain why in the text is written that “..in the context of PSP-P it is related with the evaluation of a phenotype associated with 30% cases of PSP ”while in reference n 20 (Alster P, Madetko N, Koziorowski D, Friedman A. Progressive Supranuclear Palsy-Parkinsonism Predominant (PSP-P)-A Clinical Challenge at the Boundaries of PSP and Parkinson's Disease (PD). Front Neurol. 2020 Mar 10;11:180. doi: 10.3389/fneur.2020.00180. PMID: 32218768; PMCID: PMC707866) it is written that “There is growing interest in the second most common variant of PSP—parkinsonism predominant PSP-P. It is observed in up to 35% of cases.”

Author Response

Dear Reviewer 2,

Authors are grateful for all of the comments, which improved our manuscript. We have revised the work accordingly to the suggestions:

1.Authors should describe what PSP-RS is since it is mentioned for the first time.

The change was implemented.

  1. Authors should specify if patients of the two groups are males, females or both as they done for the control group later.

The information was added.

  1. Authors should write spect in capital letters

The change was implemented.

  1. The possession administration was conducted in a supine position with….

The change was implemented.

  1. Author should explain why they are saying that “Significant results are presented as box plots”but there is no box plot in the paper.

The box plot was initially intended to be added in the supplementary material. Currently it is provided as Image 1 in the manuscript.

  1. Authors should adjust the format of the table. They can use a legend in the didascalie such as M = Median, LQ = Lower Quartile, etc. or Q1, Q3. Q3-Q1. Probably is better to add the p value of Table2 in the Table 1. It’s very difficult to correlate the data between the two tables.

The change was implemented.

  1. Authors should not highlight in red anything; they should argue the result in the text

The change was implemented.

  1. Authors should explain why is the image cut of and why there is a square only just around the pic n 63. Even more they should put also an image of the Frontal Hypoperfusion in MSA-P. It is advisable to describe the figure in more detail.

The pic on n. 63 was selected automatically and is one of the images from the sagittal plane. The images of frontal lobe in MSA-P were inserted. Additionally the cutted images were eliminated.

  1. Authors should explain why they put the data related to the whole group but they never show data related to the control group.

The analysis of the control group was based on data used in the software used in the study. The data concerning the control group is provided in the SPECT section of the manuscript:

“No control group was evaluated as part of this research, instead the data were referred to a reference database comprising the [99mTc]Tc-HMPAO brain scans of 20 healthy volunteers aged 64–86 years old (males and females). All comparisons were automatically presented as standard deviations using Scenium. The values of standard deviations from ROIs were assessed in various locations in the brain by statistical analysis. The outcome of the research was interpreted by a specialist experienced in nuclear medicine.

  1. Authors should better explain how they heve done the analysis. They wrote “The highest difference between median values of absolute SPECT perfusion between MSA-P and PSP-P patients was observed in the right middle frontal gyrus (MSA-P = 1.1 vs PSP-P = -2.9)”. Do they used the U Mann-Whitney test between which values? How the control group was used in the analysis??

Authors used U Mann-Whitney as a test to verify differentiation between MSA-P and PSP-P, as problematic entities in clinical examination. The comparison with the control group (healthy volunteers) was evaluated using the Scenium software, which provided data concerning healthy volunteers and enabled obtaining standard deviations of perfusion in the ROIs in the comparisons MSA-P/HC and PSP-P/HC.

  1. Authors should replace PSP-Richardson Syndrome by inserting the acronym PSP-RS as it will be described as PSP-Richardson Syndrome in the introduction.

The change was implemented.

  1. Generally, authors should enter acronyms or the complete name the first time they introduce a concept. See: Frontal Assessment Battery, MR, ROIs

The change was implemented.

  1. Author should replace (5,7,8,9) with (5,7-9)

The change was implemented.

  1. Authors should insert the reference when they are saying “Previous work by this research group found that mesencephalon/pons ratio and MRPI may be beneficial in the neuroimaging differentiation of PSP-P and MSA-P”.

The reference was added.

  1. Authors should explain why in the text is written that “..in the context of PSP-P it is related with the evaluation of a phenotype associated with 30% cases of PSP ”while in reference n 20 (Alster P, Madetko N, Koziorowski D, Friedman A. Progressive Supranuclear Palsy-Parkinsonism Predominant (PSP-P)-A Clinical Challenge at the Boundaries of PSP and Parkinson's Disease (PD). Front Neurol. 2020 Mar 10;11:180. doi: 10.3389/fneur.2020.00180. PMID: 32218768; PMCID: PMC707866) it is written that “There is growing interest in the second most common variant of PSP—parkinsonism predominant PSP-P. It is observed in up to 35% of cases.”

We apologise for this mistake. The information was corrected as the incidence of PSP-P is up to 35%.

Best regards,

Piotr Alster
